# Enhancing Bioactive Compound Classification through the Synergy of Fourier-Transform Infrared Spectroscopy and Advanced Machine Learning Methods

**DOI:** 10.3390/antibiotics13050428

**Published:** 2024-05-09

**Authors:** Pedro N. Sampaio, Cecília C. R. Calado

**Affiliations:** 1COPELABS—Computação e Cognição Centrada nas Pessoas, Faculty of Engineering, Lusófona University, Campo Grande, 376, 1749-024 Lisbon, Portugal; 2GREEN-IT—BioResources for Sustainability Unit, Institute of Chemical and Biological Technology António Xavier, ITQB NOVA, Av. da República, 2780-157 Oeiras, Portugal; 3ISEL—Instituto Superior de Engenharia de Lisboa, Instituto Politécnico de Lisboa, Rua Conselheiro Emídio Navarro 1, 1959-007 Lisbon, Portugal; cecilia.calado@isel.pt; 4iBB—Institute for Bioengineering and Biosciences, i4HB—The Associate Laboratory Institute for Health and Bioeconomy, IST—Instituto Superior Técnico, Universidade de Lisboa, Av. Rovisco Pais, 1049-001 Lisbon, Portugal

**Keywords:** antimicrobial, *Cynara cardunculus*, machine learning, MIR-Spectroscopy, PCA, PLS-DA, SVM, KNN, BPN

## Abstract

Bacterial infections and resistance to antibiotic drugs represent the highest challenges to public health. The search for new and promising compounds with anti-bacterial activity is a very urgent matter. To promote the development of platforms enabling the discovery of compounds with anti-bacterial activity, Fourier-Transform Mid-Infrared (FT-MIR) spectroscopy coupled with machine learning algorithms was used to predict the impact of compounds extracted from *Cynara cardunculus* against *Escherichia coli*. According to the plant tissues (seeds, dry and fresh leaves, and flowers) and the solvents used (ethanol, methanol, acetone, ethyl acetate, and water), compounds with different compositions concerning the phenol content and antioxidant and antimicrobial activities were obtained. A principal component analysis of the spectra allowed us to discriminate compounds that inhibited *E. coli* growth according to the conventional assay. The supervised classification models enabled the prediction of the compounds’ impact on *E. coli* growth, showing the following values for accuracy: 94% for partial least squares-discriminant analysis; 89% for support vector machine; 72% for k-nearest neighbors; and 100% for a backpropagation network. According to the results, the integration of FT-MIR spectroscopy with machine learning presents a high potential to promote the discovery of new compounds with antibacterial activity, thereby streamlining the drug exploratory process.

## 1. Introduction

As a consequence of the rise in antibiotic resistance, it is imperative to develop fast and affordable systems for discovering bioactive compounds with antimicrobial activity that are also non-cytotoxic to the human host [1]. Antibiotic-resistant bacteria present a formidable and concerning issue in contemporary medicine, which is compounded by the dwindling progress in new antibiotic development and the escalating spread of multi-drug-resistant determinants, exacerbating the problem. Plant extracts are important sources of biologically active compounds and are indispensable for the development and synthesis of many drugs, including antibiotics. *Cynara cardunculus* (Cardoon), for example, has been used in traditional medicine [2], namely, extracts of its rhizomes are characterized by antioxidant and antimicrobial activities [3], while its leaves show diuretic, choleretic, and hepatoprotective properties [4], in addition to in vitro anti-proliferative potential against breast cancer [5,6] and cervical cancer uterus [7]. Despite the promising potential of these plant extracts, evaluating their antimicrobial activity typically involves expensive and time-consuming techniques that demand significant sample quantities [8,9]. The current study investigates how Fourier-Transform Infrared (FTIR) spectroscopy, when coupled with machine learning (ML) and neural network modeling, can overcome these limitations and enhance its capabilities. FTIR spectroscopy, particularly in the mid-infrared region (MIR), captures the vibrational signatures of key biomolecules like proteins, carbohydrates, lipids, and nucleic acids [10]. As a result, these spectra serve as metabolic fingerprints for biological samples, offering high sensitivity and specificity [11]. This technique has been applied to classify bacteria [12], differentiate between cell death mechanisms such as apoptosis and necrosis [13], conduct studies and diagnoses related to cancer [14], monitor the effects of drugs on *Helicobacter pylori* [15,16], and characterize cell metabolism in bioreactor cultures [17,18]. This demonstrates the versatility and effectiveness of FTIR spectroscopy in diverse biomedical and biotechnological applications.

In untargeted analyses, FTIR spectra have been proven instrumental in identifying specific phenotypes. They have been utilized to uncover the antimicrobial properties of novel extracts [19], differentiate the effects of various surfactants on *E. coli* cells [20], compare how *E. coli* responds to diverse stress conditions [21], and develop bioassays for toxicity testing in yeasts [22], among other applications. Compared to various omics techniques, FTIR spectroscopy offers valuable insights with less data intensity and minimal impact on metabolic sensitivity [23]. This underscores its significance and potential for further advancements in research across different fields. Given the significant amount of information present in the obtained spectra, their analysis must be carried out using multivariate statistical methods, such as ML, in which the classification and discrimination methods used allow the development of mathematical models capable of discriminating the samples.

Machine learning (ML) is commonly categorized as a subset of artificial intelligence (AI), relying on mathematical frameworks to enable computers to learn autonomously and discern patterns and structures within both structured and unstructured datasets. This process eliminates the need for explicit programming instructions, thereby empowering machines to adapt and improve their performance over time [24]. In ML, two main types of techniques are used: supervised and unsupervised learning. Concerning unsupervised learning, hidden patterns or intrinsic structures in the input data are identified and subsequently grouped correctly [25]. Unsupervised methods usually serve exploratory purposes, aiding in the development of models that facilitate data clustering in a manner not explicitly specified by the user. Supervised learning methods are employed to construct training models that predict future values of data categories or continuous variables. Examples of supervised techniques include partial least squares-discriminant analysis (PLS-DA), k-nearest neighbors (kNNs), and support vector machines (SVMs). Other techniques include artificial neural networks (ANNs) such as backpropagation networks (BPNs) and counter-propagation networks, which are multilayer networks based on the combinations of the input, output, and clustering layers [26].

The main objective of this study was to show the advantage of FT-MIR spectroscopy and machine learning methods to screen for and, consequently, discover promissory bioactive molecules as antimicrobial agents. The extracts, obtained with different solvents, from the seeds, dry and fresh leaves, and flowers of *C. cardunculus* were evaluated. This study evaluated if the spectra of *Escherichia coli* could be used to predict the impact of rapid exposure to various compounds on cells. For this purpose, diverse unsupervised and supervised machine-learning algorithms were evaluated over the obtained *E. coli* spectral data. This approach serves as a tool in targeted exploration for novel compounds with antimicrobial activity.

## 2. Results and Discussion

### 2.1. Analysis of C. cardunculus Extracts

The compounds were extracted with diverse solvents from *C. cardunculus* flowers, seeds, and fresh and dried leaves. The following extracts presented antimicrobial activity against *E. coli*: seeds subjected to extraction with water and ethanol, leaves processed with methanol, dry leaves treated with water, and flowers subjected to extraction with ethanol and methanol (Figure 1). Previously, Ibrahim et al. [27] also observed that seed extracts showed antimicrobial activity against *Escherichia coli*, *Staphylococcus aureus*, *Staphylococcus sprophyticus*, and *Klebsiella pneumoniae* species.

Methanol, ethanol, and water were the most suitable solvents to extract compounds characterized by significant antioxidant activity. The extracts obtained from seeds showed significant antioxidant activity (AA): 75% AA for methanol and 60% AA for water. Fresh leaves also demonstrated significant antioxidant activity (76% AA for methanol and 86% AA for water). The extracts obtained from fresh leaves and seeds using a methanol and water mixture revealed substantial antioxidant activity. The antioxidant level most probably reflects the specific nature of the tissues and their respective functions in terms of plant defense. The highest phenol concentrations were registered for the aqueous extracts from flowers and seeds (8.03 µg gallic acid equivalent (GAE)/g and 3.93 µg GAE/g, respectively), most probably due to the high temperatures used during extraction (~100 °C). The extracts obtained with methanol and ethanol also presented high phenolic compounds, most probably due to the affinity of phenolic compounds with organic solvents. The extracts obtained with ethyl acetate (0.80 µg GAE/g) and acetone (1.22 µg GAE/g) presented low values, suggesting that these solvents presented low affinity for the extraction of these compounds. The present results are in accordance with the results reported by other authors, showing that extracts from *C. cardunculus* L. presented polyphenolic compounds, specifically flavonoids and phenolic acids [28]. Phenolic compounds are the main plant antioxidants and may act through their hydrogen-donor and metal-chelating capacities. Indeed, it was observed that there was a high correlation between the phenol content and antioxidant levels for the extracts obtained with ethanol (r^2^ = 0.63) and with methanol (r^2^ = 0.96). This was in accordance with Fu et al. [29], who also observed a positive correlation between the content of phenolic compounds and antioxidant activity.

The phenolic contents of plant extracts have been associated with beneficial effects on health, which include anti-allergic, anti-atherogenic, anti-inflammatory, antimicrobial, and antithrombotic effects, as well as acting as cardiovascular protectors and vasodilators [30]. The pronounced lipophilic nature of phenolic compounds enhances their antimicrobial activity, most probably through diverse mechanisms [3,5,31,32]. This antimicrobial activity can affect the permeability of cell membranes. Moreover, interactions with cell membranes can lead to changes in cell wall rigidity, resulting in a loss of integrity and ultimately inhibiting microbial proliferation [33].

### 2.2. PCA of FT-MIR Spectra of E. coli

The FT-MIR spectra of *E. coli* cells incubated with these extracts were pre-processed by different algorithms, such as multiplicative scatter correction (MSC), straight normal variate (SNV), the first derivative, and the second derivative. The impact of these spectral pre-processing methods was evaluated using PCA (Figure 2A). The most effective pre-processing methods for clustering scores in the PCA score plot according to the antimicrobial activity were MSC and the use of the second derivative. The second derivative allowed us to resolve overlapped spectral bands, while MSC minimized the effect of light scattering (Figure 2B).

The scores were clustered according to the extracts’ antimicrobial activity, as observed according to the conventional assay (based on counting colonies on agar plates) (Figure 2A,B). The clusters of the extracts with antimicrobial activity included extracts from seeds (extracted with water and ethanol), leaves (extracted with methanol), dry leaves (extracted with water), and flowers (extracted with ethanol and methanol). These extracts also presented significant antioxidant activity. According to the results obtained, 14 samples showed a residual antimicrobial activity, while 6 samples showed significant antimicrobial activity. Meanwhile, the PCA loadings highlighted the spectral regions that contributed to the resolution of scores between the two clusters characterized by high and low antimicrobial activity, especially PC2 loading, which had a strong contribution from the following regions: 1000 to 1800 cm^−1^, and 2700 to 3000 cm^−1^ (Figure 2B). The spectral regions ranging from 1600 to 1700 cm^−1^ primarily arose from amide I and amide II absorptions associated with proteins, typically constituting the most prominent bands in a bacterial infrared spectrum. This prominence is attributed to proteins accounting for approximately half of the bacterial cell composition on average. The peak registered at 1740 cm^−1^ is related to the ester C=O stretching of phospholipids. At wavenumbers around 1080 cm^−1^ and 1240 cm^−1^, there is a significant presence of the P=O bond characteristic of phosphate groups found in molecules such as nucleotides (like ATP), nucleic acids, phosphorylated proteins, and lipids. Between 1000 and 1100 cm^−1^, there is a broad contribution from bonds such as C–O, C–C, C–OH, and C–O–C, which are commonly found in carbohydrates. The spectral region between 2700 and 3000 cm^−1^ exhibits a notable intensity in the stretching vibrations of C-H bonds, particularly from lipids. This resonance is particularly pronounced in bacterial membranes (as outlined in Table 1).

### 2.3. Analysis of Spectral Bands

The spectral second derivative presents bands that, on average, are different between *E. coli* subjected to compounds with antimicrobial or low antimicrobial activity (Figure 3A,B). To further identify spectral bands that exhibit significant differences between bacteria exposed to inhibitory and non-inhibitory compounds, several ratios between the bands, as highlighted by the PCA loading vector, were examined. Twelve ratios between the spectral bands were analyzed, where “A” followed by a number represents the absorbance at that wavenumber (Table 2). From the twelve ratios evaluated, the following nine ratios were statistically different at the 5% significance level between the two clusters: A2856/A1705, A1740/A1656, A1740/1545, A2847/1545, A1617/A1545, A1476/1545, 1215/1179, A1215/1545, and A1244/1230. These ratios reflect the compounds’ impact on bacterial metabolism related to protein synthesis (as indicated by the ratio between amides I and II, i.e., A1617/A1545, and various ratios involving the amide I peak). Additionally, changes in the composition of the cell membrane are highlighted by the CH_3_/CH_2_ ratio (i.e., A2912/A2856), as well as the ratios incorporating the bands at 1740 cm^−1^ and 1179 cm^−1^ corresponding to carbonyl vibrations of esters of phospholipids, and the bands at 2856 cm^−1^ and 1476 cm^−1^ originating from methyl and methylene groups. Figure 4 presents the box plots for the ratios of spectral bands that are statistically different. These findings underscore the remarkable sensitivity of FT-MIR spectroscopy in capturing the influence of compounds on *E. coli* metabolism.

### 2.4. Classification Models

To assess the feasibility of predicting the impact of compounds on *E. coli* inhibition, based on the spectral data from bacterial cells, several supervised classification machine learning algorithms were evaluated, including PLS-DA, k-NN, SVM, and BPN. For all models, 70% and 30% of the spectra were used as the training data (*n* = 42) and test data (*n* = 18), respectively (Table 2). The calibration and cross-validation performances of the tested models were compared in terms of classification parameters, such as non-error rate (NER) and accuracy, with an analysis of confusion matrices, thus providing a comprehensive overview and a rational means of selecting the best model.

#### 2.4.1. PLS-DA Model

The most appropriate PLS-DA model was developed through the implementation of the spectra after MSC pre-processing, explaining a total variance of 99% for 11 LVs (Table 3). The error for the validation procedure was 11%. The model presented sensibility, specificity, precision, and accuracy of 86%, 92%, 96%, and 88%, respectively.

The PLS-DA model’s performance could be affected by outliers, as predicted by several parameters such as leverages, Q residuals, and Hotelling’s T2. The samples exhibited low residual Q and Hotelling’s T2 values, suggesting minimal variance between the actual and predicted outputs of the model [34]. Figure 5A illustrates the response of cluster A (exhibiting antimicrobial activity) for the training samples. The classification threshold to distinguish cluster A (antimicrobial activity) from cluster B (low antimicrobial activity) was set at 0.210 (depicted by the dotted red line).

Misclassified samples were evident in our analysis, with samples from cluster A being mistakenly categorized in cluster B (false negatives), and samples from cluster B erroneously assigned to cluster A (false positives) (Figure 5A). The score plot indicates a discernible pattern in the distribution of samples, with a noticeable degree of overlap between them. The first latent variable explains 39% of the variance, suggesting it captures a significant portion of the underlying structure in the data. However, the second latent variable explains only 12% of the variance, indicating that there is still substantial variability not accounted for by the model (Figure 5B). In Figure 5B, it is evident that samples belonging to cluster A, which are characterized by antimicrobial action, exhibit higher scores for the first latent variable (LV) compared to samples from cluster B, which have low antimicrobial activity. This suggests that the first LV effectively discriminates between the two clusters based on their antimicrobial properties. Concerning the second LV, samples from cluster B are represented by negative scores, while samples from cluster A exhibit positive scores. This distinction demonstrates a significant resolution between the two types of plant extracts, further highlighting the efficacy of the model in distinguishing between them based on their characteristics. In the validation step, it was observed that the samples utilized in the test phase for both groups were placed close to the samples utilized in the training set. This finding indicates that the model exhibits high accuracy, thereby adding value to the classification process facilitated by the PLS-DA technique. The studies conducted by Pellegrini et al. (2017) showed the advantages of PLS-DA models as classification models for different essential oils extracted from *B. latifolia*, *Buddleja globosa*, *Solidago chilensis*, and *Aloysia polystachia,* which showed antimicrobial activity against *Paenibacillus larvae* [35].

#### 2.4.2. K-Nearest Neighbor (KNN)

The KNN-optimized models, developed based on the spectra with MSC pre-processing, showed a high accuracy value (90%) and a low classification error (12%), with a cross-validation error rate of 20% (Table 3). The training model was characterized by an error of 13%, and sensitivity, specificity, and accuracy of 90%, 85%, and 88%, respectively, in predicting antimicrobial activity. For the cross-validation dataset, we obtained an error of 10% and sensitivity and specificity of 92% and 90%, respectively, in predicting antimicrobial activity (Table 3). For the test dataset, the accuracy, specificity, and sensitivity in predicting antimicrobial activity were 72%, 100%, and 0%, respectively. The performance of KNN revealed that for the classification of samples characterized by antimicrobial activity, the classification rate was 88% and 90% for the training and validation process.

#### 2.4.3. Support Vector Machine (SVM)

The SVM model, based on the *E. coli* spectra with MSC pre-processing, was developed after the fine-tuning of several parameters and was characterized by a radial basis function SVM-kernel, with C = 100. The model showed significant fitting accuracy (90%), with a 12% classification error and a 20% cross-validation error rate (Table 3). The training model was characterized by excellent accuracy (93%) and a low calibration error (9%). The sensitivity and specificity for cluster A (antimicrobial activity) were 97% and 85%, respectively. In terms of the calibration model, the specificity and sensitivity for cluster A were 80% and 85%, respectively. In terms of cross-validation, the accuracy and validation error were, respectively, 81% and 20%, being characterized by excellent specificity and sensitivity (92% and 93%) in predicting cluster A (Table 3). In terms of prediction, the model presented an accuracy value of 89%, with values of 80% and 85% for specificity and sensitivity, respectively, in predict clustering A. The SVM model combined with the MIR spectral data, therefore, was a robust model for predicting antimicrobial activity.

#### 2.4.4. Backpropagation Network (BPN)

The success of an artificial neural network depends on the size and quality of the training data, as well as its structure and the learning algorithms used. The samples were randomly assigned to a training set (65%) and a test set (35%). To optimize the network, an appropriate learning rate and different numbers of input nodes (10 and 20) and hidden layer neurons (1 and 2) were evaluated. According to the test set values obtained for the different models, the BPN algorithms (1:10; 1:20; 2:10, and 2:20) presented the best results when compared to the other algorithms (Table 3). Based on the MSE value, the model developed with one layer and 20 nodes was more suitable for the classification process. For each condition, 20 neurons per layer were considered, with a learning rate = 0.01, 1000 iterations, and a momentum term of 0.5, which allowed for faster convergence with the use of smaller learning coefficients. A two-layer BP model was developed at last, with the sigmoid transfer function. The scores of the first 15 PCs were applied as the input variables in the BP-ANN models. The epoch number was set at 1000, and after the multi-training step, the optimal number of nodes in the hidden layer and the learning rate were defined as 20 and 0.01, respectively. The optimized BPN model, based on the spectra with MSC pre-processing, showed an excellent fitting accuracy of 90%, a low classification error of 12%, and a cross-validation error rate of 20% (Table 3). The training model was characterized by an accuracy of 100% and a calibration error of 22%. Sensitivity was 100% and specificity was 100% in predicting the antimicrobial effect (Table 3). For the test dataset, the error rate was 14% and the accuracy value was 89%, showing sensitivity of 92% and specificity of 80% for cluster A.

The BPN 1:20 model was the one with the best classification results in terms of the test conditions, followed by the BPN model with two layers and 10 nodes and the BPN model with two layers and 20 nodes. Based on these results, the BPN algorithms provide the best guarantee for the development of classification models to predict extracts with antimicrobial activity (Figure 6). These neural networks have an advantage over conventional mathematical methods for modeling complex biological systems because (1) they are based only on an actual measured set of input and output variables, without requiring prior information about the interrelationship between the variables, and (2) they possess strong generalization and prediction ability. The principal advantages of artificial neural network (ANN) techniques include their capacity for data learning and generalization, fault tolerance, and an innate ability to process contextual information, alongside their rapid computational capabilities [36]. According to Badura et al. (2021), ANNs serve as excellent tools to support researchers in laboratory work, enabling the estimation of whether a tested compound possesses the desired antimicrobial activity [37]. It is also noteworthy that the primary advantage of ANNs is their ability to streamline the work of researchers who aim to uncover the intricate relationships among the physicochemical properties of compounds. This approach enables researchers to target the chemical synthesis of compounds with the desired activity by leveraging theoretical models obtained through computational chemistry [37]. According to the studies conducted by Cabrera et al. (2010), artificial neural networks serve as reliable, fast, and cost-effective tools, paving the way for predicting the antioxidant activity of essential oils [38].

## 3. Materials and Methods

### 3.1. Extraction Process

The plant materials, including the flowers, seeds, and fresh and dried leaves of *C. cardunculus*, were subjected to maceration using a mortar and pestle. Subsequently, the macerated materials were transferred to a flask containing the respective solvent (water, ethanol, acetone, methanol, and ethyl acetate) at a concentration of 5% (*w*/*v*). The mixture was stirred at room temperature for 16 h. The extracts were then filtered under vacuum, and the solvents were evaporated using a rotavapor at 60 °C. The resulting dry extract was weighed, and a suitable volume of H_2_O was added to achieve a final concentration of 10 mg/mL. In the case of aqueous extraction, the material was boiled at approximately 100 °C for 10 min (Figure 7).

### 3.2. Antioxidant Activity

The antioxidant activity was determined according to the procedure developed by Mensor et al. [39], with slight adaptations as follows: the samples were diluted in methanol at concentrations of 10, 50, and 200 µg/mL. A total of 1 mL of alcoholic solution of 2,2-diphenyl-1-picryl-hydrazyl (DPPH) (50 µg/mL) was added to 2.5 mL of the sample. The mixture was maintained in the dark for 30 min. Absorbance was evaluated using a spectrophotometer at 518 nm (Shimadzu, Japan). The percentage of free radical scavenging (%FRS) was evaluated according to Equation (1): *Ac*–control absorbance; *As*–sample absorbance. The blank was prepared by replacing DPPH with ethanol in the reaction mixture. The negative control was prepared by adding 1 mL of DPPH in 2.5 mL of ethanol.
(1)$$FRS\;\left(\%\right)=\frac{\left(Ac-As\right)}{Ac}\times 100$$

### 3.3. Total Phenols

The spectrophotometric determination of phenolic compounds was conducted according to the method developed by Slinkard and Singleton [40]. The calibration curve was obtained using different concentrations of gallic acid (0–1000 mg/L). Each sample (1 mL) was transferred to a 25 mL flask, while the blank was prepared with distilled H_2_O. Subsequently, 1 mL of Folin–Ciocalteu reagent was added, and the mixture was stirred and then left to rest for 5 min. To this mixture, 7 mL of (0.2 g/L) Na_2_CO_3_ solution was added, and the resulting mixture was kept in the dark for 2 h. The absorbance of the samples was determined at 765 nm, and the values were consequently converted to gallic acid equivalent (GAE) per gram of sample.

### 3.4. Antimicrobial Activity

The extracts obtained from different tissues of *C. cardunculus,* using diverse solvents, were tested on agar plates against *E. coli* cells. The bacteria were grown at 37° C with a stirrer (200 rpm) for 16 h. Then, 100 µL of the extracts was incubated with a similar volume of culture medium containing a specific concentration of cells. After an incubation period (2 h at 37 °C), 100 µL of this mixture was plated on a solid culture medium containing 2% (*w*/*v*) of agar. The plates were incubated at 37 °C for 16 h before counting the number of colonies.

### 3.5. FTIR Spectral Analysis

#### 3.5.1. Acquisition of Spectra

The samples containing *E. coli* cells were subsequently diluted to obtain a similar optical density (OD = 0.2) for each assay. All assays were performed in triplicate. The plant extracts (100 µL) were incubated with the cells in the culture medium for 2 h at 37 °C. After that, the microtubes were centrifuged at 10,000 rpm for 3 min. The pellet was resuspended in 100 µL of 0.9% (*w*/*v*) NaCl. Then, 30 µL of these final suspensions was pipetted, in triplicate, into Zn-Se plates and, subsequently, dried under a vacuum system for 3.5 h at room temperature. The spectra were obtained with an HTS-XT system (Vertex, Bruker Optics, Mannheim, Germany) equipped with an FTIR spectrometer (Bruker Optics, Mannheim, Germany) between 4000 and 400 cm^−1^ with a resolution of 4 cm^−1^ (Figure 7).

#### 3.5.2. Spectral Pre-Processing

The spectra were pre-processed by baseline correction, multiplicative scatter correction (MSC), standard normal variate (SNV), and derivatives (1st and 2nd derivatives).

### 3.6. Chemometric Methods

#### Principal Component Analysis

The compression and classification of data can be performed using the principal component analysis (PCA) technique. The goal is to reduce the dimensionality of a dataset (sample) by identifying a new set of variables that is smaller than the original set but still retains most of the information. This method groups the samples according to the existing information so that individuals in a group are as similar to each other as possible and as different from the remaining groups as possible (Figure 7).

### 3.7. Classification Methods

#### 3.7.1. Partial Least Squares-Discriminant Analysis (PLS-DA)

Partial least squares-discriminant analysis (PLS-DA) is a linear classification tool used to calculate predictive models. It employs the partial least squares regression algorithm to identify latent variables that exhibit maximum covariance between the predictor variables and the class labels [34,41]. The PLS-DA models were calibrated using a total of 42 samples that were randomly selected: 29 samples with antimicrobial effect and 13 samples with no antimicrobial effect (Figure 7). Cross-validation was conducted using randomly selected training samples, with 5 data splits comprising 20% of the calibration samples each. This process was repeated for 20 iterations to ensure the robustness and reliability of the results. The test samples were constituted by 18 samples that were randomly selected: 13 samples with antimicrobial effect and 5 samples with no antimicrobial effect. Prediction was performed with the samples of the testing set. The evaluation of the prediction ability of the models was conducted based on their sensitivity (*Sn*), specificity (*Sp*), and accuracy, defined according to the true positives (*tp*), true negatives (*tn*), false positives (*fp*), and false negatives (*fn*), as follows:

Sensitivity (*Sn*):(2)$$Sn=\frac{tp}{tp+fp}\times 100\%$$

Specificity (*Sp*):(3)$$Sp=\frac{tn}{tn+fp}\times 100\%$$

Accuracy (*Acc*):(4)$$Acc=\frac{tp+tn}{tp+fn+tn+fp}\times 100\%$$

#### 3.7.2. K-Nearest Neighbors

The K-nearest neighbors (KNN) algorithm is a supervised machine learning algorithm in which classification rules are derived from the neighborhood of training-set objects. Here, ‘k’ represents the number of neighbors surrounding the new data point, and classification is based on the majority class among its nearest neighbors. Distance functions play a crucial role in computing the distance between the new data point and its ‘k’ neighbors [42]. The neighborhoods are determined by calculating either Euclidean distances or correlation coefficients between the unknown object and the objects in the training set. For a training set comprising *n* samples, *n* distances or correlations are computed. The unknown object is assigned to the class to which the majority of the *k* objects with the smallest Euclidean distances or the highest correlations among the training samples belong.

#### 3.7.3. Support Vector Machine

Support vector machine (SVM) relies on identifying a hyperplane that effectively divides a dataset into two distinct groups. However, when dealing with non-linear data, SVM may struggle with separating the data using linear hyperplanes. In such cases, alternative methods, known as kernel functions, are employed. Kernel functions are mathematical functions that take data as input and transform them into a higher-dimensional space, allowing SVM to effectively classify non-linear data by finding the optimal separating hyperplane in the transformed feature space [43]. Kernel functions map data to a higher dimension, enabling SVM classifiers to achieve optimal performance. The parameters and kernels are fine-tuned to enhance SVM classifier performance. The regularization parameter ‘C’ balances between minimizing training error and model complexity. Additionally, the kernel parameter ‘γ’ determines the width of the kernel function and the degree of generalization. These parameters are optimized using a grid-search procedure in SVM. In this study, the optimal conditions were determined by testing a regularization parameter ‘C’ value of 100 and utilizing a linear kernel function.

#### 3.7.4. Backpropagation Network (BPN)

A backpropagation network (BPN) is highly regarded in data processing due to its ability to model non-linear processes, leverage data-driven features for powerful parallel computing, and demonstrate good fault tolerance and adaptability [44]. A backpropagation network (BPN) utilizes a gradient descent-based delta learning rule, commonly known as backpropagation, to train an artificial neural network [45]. This systematic method efficiently adjusts the weights in the network with functional units to analyze a set of input–output patterns. It aims to minimize the total squared error of the output, thus enabling the trained supervised learning network to effectively balance its ability to accurately respond to input patterns. A BPN algorithm minimizes prediction errors and yields satisfactory results by adjusting each weight of the network. This method is employed when establishing an artificial neural network (ANN) model [46]. During the optimization step, the BPN consisted of either 10 or 20 neurons distributed across one or two hidden layers. In the modeling process, 70% of the total samples were allocated and utilized as the training dataset for constructing the model, comprising 42 samples. To mitigate the risk of overfitting, 30% of the samples were set aside and employed as the validation dataset for implementing the early stopping methodology, consisting of 18 samples. Based on the preliminary experiment, the divider and function were selected to randomly partition the sample data in this study. The Mean Square Error (*MSE*) serves as a commonly used metric in the field of machine learning and neural networks, including BPNs. *MSE* is a measure of the average squared difference between the actual and predicted values. In the context of a backpropagation network (BPN) model during the training process, *MSE* quantifies the extent to which the model’s predictions deviate from the actual target values. The formula for *MSE* is typically expressed as shown in Equation (5):(5)$$MSE=\frac{1}{N}\sum_{i=1}^{N}{\left(y_{i}-{\hat{y}}_{i}\right)}^{2}$$
where *N* is the number of data points in the dataset; $y_{i}$ represents the actual target value for the ith data point; and ${\hat{y}}_{i}.$*MSE* is calculated by taking the average of the squared difference between the actual and predicted values for all data points. The use of the squared difference helps to penalize larger errors more significantly than smaller errors. A BPN neural network is a kind of multilayer feed-forward neural network. The specific topological structure of the BP neural network is shown in Figure 8. The BP-ANN model was developed using the ANN tool of MATLAB 2023a software.

Neurons are organized into layers within a neural network. The layers situated between the input layer, where the input signal is applied, and the output layer, which generates the output signal, are referred to as the hidden layers. The input layer comprises a total of 1154 absorbances of the entire spectrum. Each hidden layer consists of 20 nodes. The output layer represents the classification results of the extracts, which were categorized as either having antimicrobial activity or no antimicrobial activity. During the training process, the network learns each time a training pattern (comprising input and output data) is presented to it. This learning occurs through the modification of connection weights, and the process is repeated several times (iterations or epochs) until a satisfactory level of error is attained. The MLP parameters were trained using backpropagation to minimize the MSE, thereby optimizing the weights and biases for each neuron. Subsequently, the output layer yielded the result of the hidden layers, which was indicative of the antimicrobial activity. The training process was conducted with a learning rate of 0.001 for 500 epochs, each comprising 1000 batches. Each batch contained 1154 absorbances. At each epoch, the MLP was evaluated using a validation set of the same size as the training set. The MLP parameters corresponding to each epoch were recorded.

### 3.8. Other Statistical Analysis

Student’s *t*-test was performed using a data analysis tool package (Microsoft Office 365 Excel^®^) to evaluate differences in the ratios between the bands, at a 5% significance level. Spectral pre-processing and processing were conducted with a classification toolbox (PLS-DA, SVM, k-NN, and BPN) developed by Milano Chemometrics and QSAR Research Group (http://michem.disat.unimib.it/chm (accessed on 1 September 2023)), using MATLAB^®^ 2023a (The MathWorks, Natick, MA, USA).

## 4. Conclusions

The results obtained in this study suggest that the integration of FT-MIR spectroscopy with machine learning methods represents a promising strategy to evaluate the impact of biocompounds obtained from plant extracts as antimicrobial agents against bacterial cells. This holistic approach holds the potential for efficiently classifying and discovering new bioactive molecules, thereby streamlining the drug exploratory process. It is possible to develop a model based on a backpropagation network for plant extract classification based on the spectra of *E. coli* after rapid exposure to antimicrobial compounds. The combination of FTIR spectroscopy and machine learning techniques creates the opportunity for preliminary research of promising antimicrobial compounds, saving time and resources compared to tests that require greater logistics in terms of consumables and laboratory techniques. Data analysis employing artificial neural networks offers a powerful method to optimize and minimize labor costs by streamlining the synthesis process, focusing only on compounds with anticipated desired properties. This approach serves as an invaluable tool in guiding the design of synthesis procedures and subsequent biological experiments, particularly in the targeted exploration for novel compounds with significant potential as antimicrobial agents.

## Figures and Tables

**Figure 1 antibiotics-13-00428-f001:**
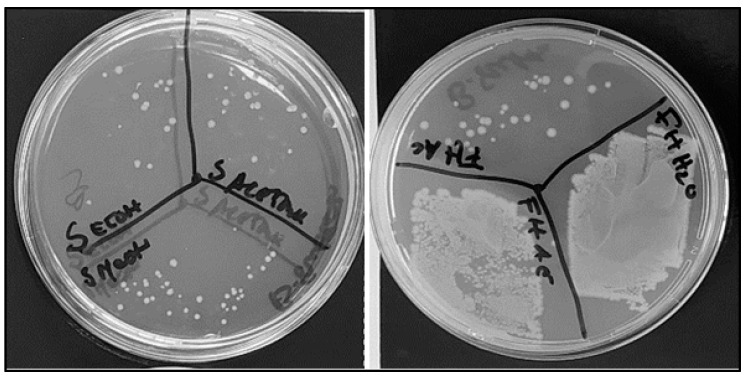
Antimicrobial activity of different assays using *C. cardunculus* extracts in *E. coli* cells in solid culture medium: seeds extracted with ethanol; seeds extracted with methanol; seeds extracted with acetone; fresh leaves extracted with acetone; fresh leaves extracted with ethyl acetate; and fresh leaves extrated with water.

**Figure 2 antibiotics-13-00428-f002:**
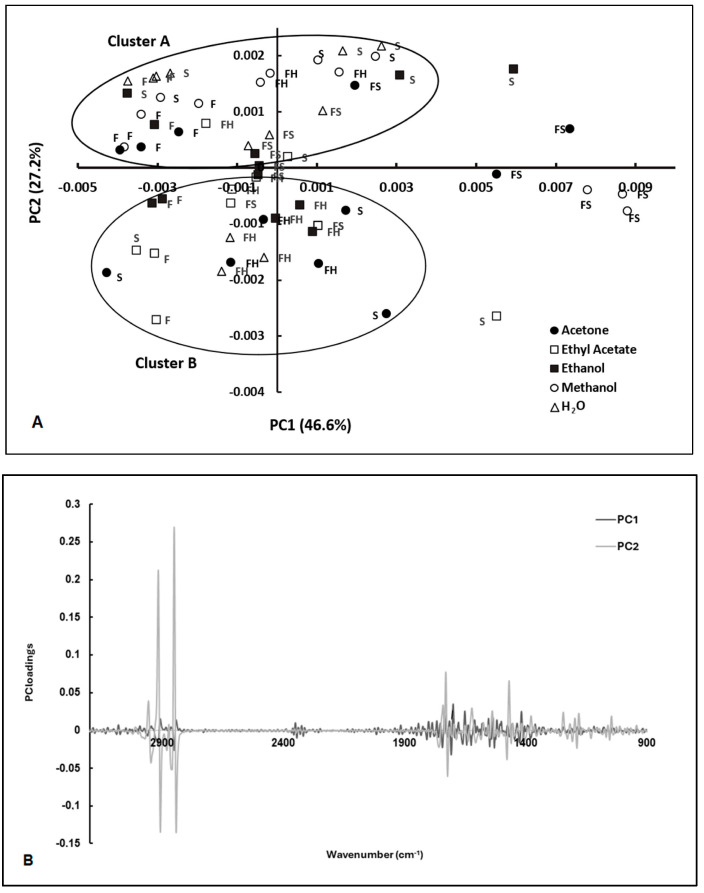
PCA of FT-MIR spectra of *E. coli* exposed to extracts from *C. cardunculus* tissues (S—seeds; F—flowers; FH—fresh leaves; FS—dry leaves) obtained by different solvents (AE—ethyl acetate, EtOH—ethanol; Ac—acetone; MeOH—methanol, and H_2_O—water) (**A**), and the corresponding loading vectors (**B**). Spectra were pre-processed by MSC and 2nd derivative. Cluster A is defined by samples characterized by antimicrobial activity. Cluster B is constituted by samples characterized by low antimicrobial activity.

**Figure 3 antibiotics-13-00428-f003:**
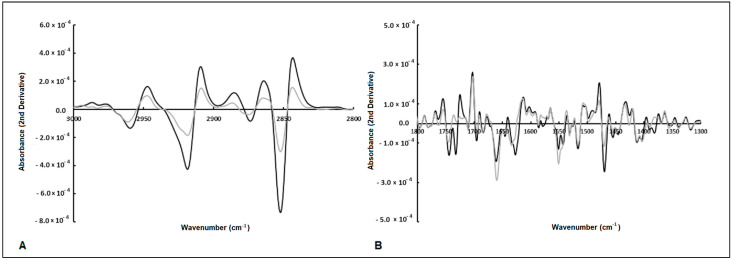
Average of second-derivative spectra of *E. coli* exposed to antimicrobial (gray line) and non-antimicrobial (black line) compounds. The arrows highlight spectral bands that are different between the two bacterial populations (**A**,**B**).

**Figure 4 antibiotics-13-00428-f004:**
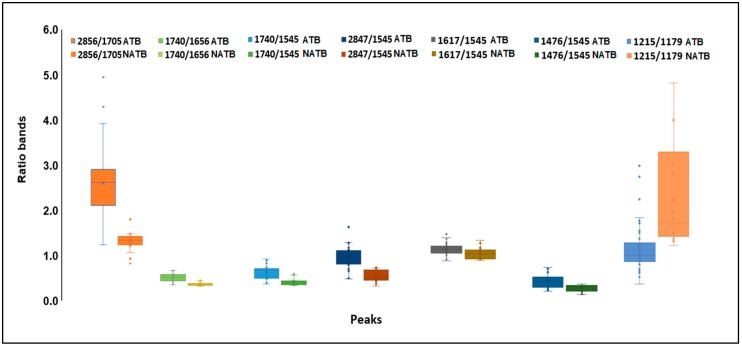
Box plots of ratios of spectral bands of *E. coli* cells exposed to compounds with (A) and without (NA) inhibitory activities.

**Figure 5 antibiotics-13-00428-f005:**
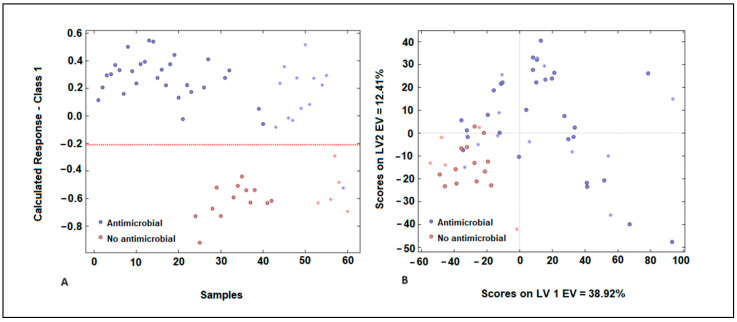
Representation of samples in the group with antimicrobial activity (blue) and the group without antimicrobial activity (red) of plant extracts (**A**) for the PLS-DA model developed using *E. coli* spectra with MSC pre-processing. Score plot obtained during the development of the PLS-DA model (**B**).

**Figure 6 antibiotics-13-00428-f006:**
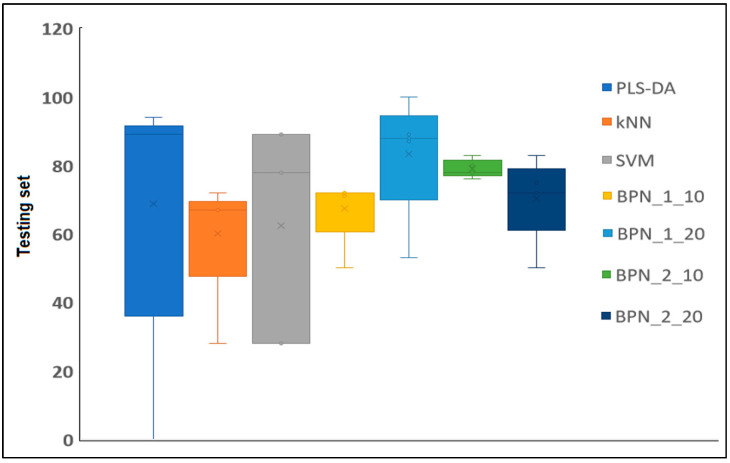
Comparison of test parameters related to different classification methods being studied (PLS-DA, kNN, SVM, BPN:1:10; BPN:1:20; BPN:2:10; and BPN:2:20 models).

**Figure 7 antibiotics-13-00428-f007:**
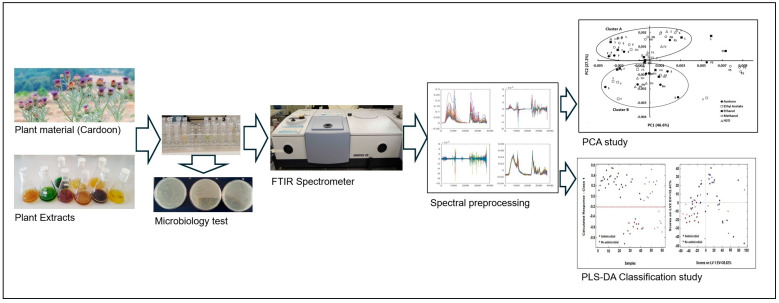
Experimental procedure carried out from selecting the plant material to extract classification based on *E. coli* FTIR spectra and machine learning methodology.

**Figure 8 antibiotics-13-00428-f008:**
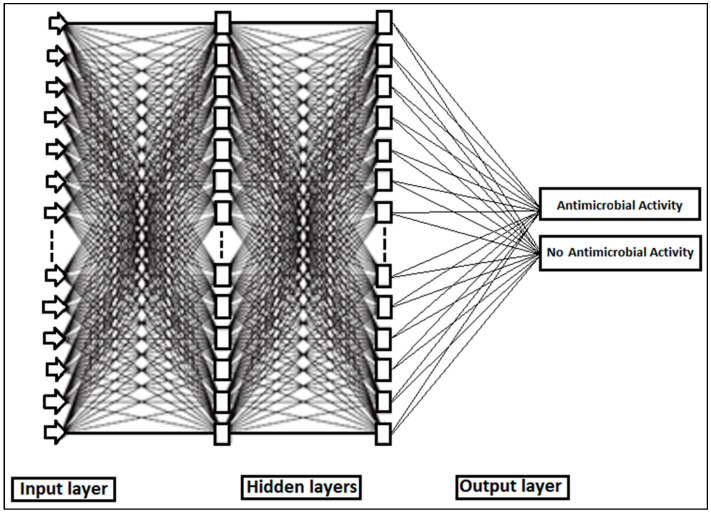
Topological structure of neural network used for the identification of antimicrobial activity.

**Table 1 antibiotics-13-00428-t001:** Vibrational frequencies of some functional groups in biomolecules (adapted from Sales et al. [17]).

Wavenumber (cm^−1^)	Vibrational Mode in the Functional Group	Commonly Assigned Biochemical Component
3300	ν(N-H)	Amide A: peptides, proteins
3100	ν(N-H)	Amide B: peptides, proteins
2960	ν_as_(CH_3_)	Lipids
2929 2870 2850	ν_as_(CH_2_) ν_s_(CH_3_) ν_s_(CH_2_)	Lipids
1740	ν(C=O)	Phospholipid esters
1655	[80% ν(C=O) and 20% ν(CN)] in O=C-N	Amide I in peptides and proteins
1550	ν(C-C) in O=C-N	Amide II in peptides and proteins
1450	ν_as_(CH_3_)	Lipids, proteins
1395	ν_s_(CH_3_)	Lipids, proteins
1380	ν_s_(CH_3_)	Phospholipids, fatty acids, triglyceride
1240	ν_s_(PO_2_^−^)	Phosphodiesters in nucleic acids, phospholipids, and phosphorylated proteins
1160 & 1111	ν_s_(C-O)	Ribose in RNA
1078	ν_s_(PO_2_^−^)	Phosphodiesters
1055	ν_s_(C-O-P)	Phosphate esters
1032	def(C-OH)	Glycogen

Vibration type: ν_s—_symmetric stretching_;_ ν_as—_asymmetric stretching_;_ def—deformations.

**Table 2 antibiotics-13-00428-t002:** Ratios between spectral bands of *E. coli* exposed to biocompounds with and without antimicrobial activities.

Ratio between Bands (cm^−1^)	Average		Standard Deviation		*p*-Value
	Antimicrobial	Non-Antimicrobial	Antimicrobial	Non-Antimicrobial	
A2912/2856	1.359	1.356	0.114	0.162	0.472
A2912/1740	3.921	4.121	1.086	1.114	0.263
A2856/1705	2.601	1.334	0.753	0.248	**<0.0001**
A1740/1656	0.261	0.114	0.089	0.038	**<0.0001**
A1740/1545	0.434	0.218	0.151	0.082	**<0.0001**
A2847/1545	0.977	0.537	0.253	0.124	**<0.0001**
A2847/1740	2.395	2.720	0.657	0.966	0.102
A1617/1545	1.129	1.035	0.124	0.131	**0.008**
A1476/1545	0.304	0.146	0.167	0.081	**<0.0001**
A1215/1179	1.269	2.950	0.605	2.314	**0.004**
A1215/1545	0.942	0.708	0.174	0.082	**<0.0001**
A1244/1230	1.312	1.158	0.180	0.027	**<0.0001**

**Table 3 antibiotics-13-00428-t003:** Performance of the prediction models of the impact of compounds on inhibiting *E. coli*, based on the FT-MIR spectra of *E. coli*, after diverse pre-processing methods. Data are shown for the training and cross-validation dataset.

Model	Calibration			Cross-Validation			Test		
	NER	ER	Accuracy	NER	ER	Accuracy	NER	ER	Accuracy
PLS-DA	100	0	100	88	22	86	92	8	89
PLS-DA msc	100	0	100	89	11	88	96	4	94
PLS-DA snv	97	3	95	89	11	88	92	8	89
PLS-DA 1st	100	0	100	78	22	81	50	50	72
PLS-DA 2nd	100	0	100	50	50	100	0	100	0
kNN	70	30	76	69	31	69	52	48	67
kNN msc	82	18	83	93	7	90	65	35	67
kNN snv	79	21	83	85	15	86	65	35	67
kNN 1st	87	13	88	91	9	90	50	50	72
kNN 2nd	74	26	76	69	31	71	50	50	28
SVM	77	23	83	58	42	55	72	28	78
SVM msc	91	9	93	80	20	81	92	8	89
SVM snv	87	13	90	80	20	81	92	8	89
SVM 1st	92	8	95	79	21	83	50	50	28
SVM 2nd	100	0	100	68	32	79	50	50	28
BPN:1:10	100	0	100	60	40	62	68	32	72
BPN:1:10 msc	96	4	95	90	10	90	81	19	71
BPN:1:10 snv	98	2	98	77	23	79	68	32	72
BPN:1:10 1st	100	0	100	73	27	80	62	38	72
BPN:1:10 2nd	85	15	87	51	49	52	47	53	50
BPN:1:20	100	0	100	68	32	71	100	0	100
BPN:1:20 msc	100	0	100	78	22	81	86	14	89
BPN:1:20 snv	100	0	100	86	14	86	81	19	87
BPN:1:20 1st	100	0	100	60	40	74	67	33	88
BPN:1:20 2nd	96	4	94	63	37	69	43	57	53
BPN:2:10	98	2	97	74	26	79	82	18	83
BPN:2:10 msc	98	2	98	86	14	87	78	22	76
BPN:2:10 snv	100	0	100	77	23	80	72	28	78
BPN:2:10 1st	84	16	85	72	25	82	75	25	80
BPN:2:10 2nd	100	0	100	62	38	60	66	34	78
BPN:2:20	100	0	100	60	40	64	50	50	72
BPN:2:20 msc	100	0	87	87	13	88	82	18	83
BPN:2:20 snv	100	0	100	72	28	78	68	32	72
BPN:2:20 1st	100	0	100	53	47	58	58	42	75
BPN:2:20 2nd	100	0	100	58	42	68	53	47	50

NER—non-error rate; ER—error rate.

## Data Availability

Data are contained within the article.

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
