# Peer review of "Enhancing Bioactive Compound Classification through the Synergy of Fourier-Transform Infrared Spectroscopy and Advanced Machine Learning Methods"

_antibiotics, 2024, doi:10.3390/antibiotics13050428_

Round 1

Reviewer 1 Report

Comments and Suggestions for Authors

The manuscript entitled "Enhancing Bioactive Compound Classification through the Synergy of Fourier-Transform Infrared Spectroscopy and Advanced Machine Learning Methods" elucidates the application of FT-MIR spectroscopy and machine learning techniques for the purpose of screening and subsequently identifying potential bioactive compounds with

antimicrobial properties. The manuscript is interesting and eventually contains relevant information for readers, However, there are few points to be addressed by the authors. The technique is novel, and because to my lack of familiarity with it, I will refrain from providing detailed commentary.

-The authors stated The spectral regions between1500 and 1800 are mainly due to amide I and amide II adsorptions related to the proteins. The frequencies of Amide I C=O is between 1600 and 1700 cm -1 . Please correct.

- The authors claim that FTMIR spectroscopy and machine learning techniques create the opportunity for preliminary research of promising antimicrobial compounds, saving time and resources compared to tests that require greater logistics in terms of consumables and laboratory techniques. Could you explain how?

I think FTMIR is a good technique that confirm the antibacterial of the extract, but is not providing us with information related to the specific compound in the mixture that causes the inhibition.

Author Response

Manuscript ID: antibiotics-2965304

Title: Enhancing Bioactive Compound Classification through the Synergy of Fourier-Transform Infrared Spectroscopy and Advanced Machine Learning Methods

Dear Editor,

We would like to thank you for the opportunity to improve and clarify the information present in the manuscript.  Then we resent, for your consideration, the revised version of our manuscript entitled “Enhancing Bioactive Compound Classification through the Synergy of Fourier-Transform Infrared Spectroscopy and Advanced Machine Learning Methods” that we would like to see published in the Special Issue Microbial Natural Products as a Source of Novel Antimicrobials (Antibiotics), that was significantly improved according to the referee’ suggestions. The manuscript was completely reviewed, and the main changes performed in the manuscript are highlighted in yellow marks.

I would like to inform you that the affiliation of the author Cecília Calado must be updated. Please, consider the following information:

4iBB – Institute for Bioengineering and Biosciences, i4HB – The Associate Laboratory Institute for Health and Bioeconomy, IST – Instituto Superior Técnico, Universidade de Lisboa, Av. Rovisco Pais, 1049-001 Lisbon, Portugal.

            I would like to inform you that all authors agree to publish this work in your journal.

Yours sincerely,

Pedro Sousa Sampaio

Reviewer 1

Dear Reviewer,

We would like to thank you for the opportunity to improve and clarify the information present in the manuscript.  

Comment 1: The authors stated The spectral regions between 1500 and 1800 are mainly due to amide I and amide II adsorptions related to the proteins. The frequencies of Amide I (C=O) are between 1600 and 1700 cm-1. Please correct.

Answer: Thank you for your suggestion. The wavenumber values were corrected:

“The spectral regions between 1600 and 1700 cm-1 are mainly due to amide I and amide II adsorptions related to the proteins and, usually, represent the most prominent bands in a bacterial infrared spectrum since proteins tend to be, on average, half of bacterial cell composition.”

Comment 2: The authors claim that FT-MIR spectroscopy and machine learning techniques create the opportunity for preliminary research of promising antimicrobial compounds, saving time and resources compared to tests that require greater logistics in terms of consumables and laboratory techniques. Could you explain how?

Answer: Thank you for your suggestion. The conclusion is in accordance with other studies that show the advantage of the FT-MIR techniques comparatively with the classical methods.

Based on the recent review, published by Jafari et al. (2024), studies performed previously showed that traditional methods have limitations, such as longer incubation times, qualitative results, in vitro-in vivo discordance, and lack of reproducibility. Advanced techniques, such as MALDI-TOF/MS, as well as electronic and vibrational techniques, such as fluorescence, FT-IR, and Raman spectroscopies, provide rapid, sensitive, qualitative, and quantitative results in a shorter time and can detect the microbial phenotypes in conditions that are closer to their in vivo environments.

According to Ranzam et al. (2024), it is stated that FT-IR technique is considered a rapid, non-destructive, inexpensive, and high-throughput analytical tool. In the same work, it is related that classical methods require about 48 h to determine their susceptibility to antibiotics, Abu-Aqi et al. isolated 1789 isolates of E. coli directly from patients’ urine measured by FT-IR microscopy and analyzed through Machine Learning algorithms (MLAs). The investigation reported that samples of urine were identified and susceptibility was tested within 40 min of being received. (Abdu-Aqil et al. 2022).

Studies performed by Suleiman et al. (2022) showed the potential of FTIR spectroscopy together with machine learning algorithms, to determine the susceptibility of P. aeruginosa to different antibiotics in a period of 20 min after the first culture. Other studies, carried out by Jafari et al. (2024) showed also that the spectroscopic diagnostic technique, allowed to reduce detection time and to improve quality in the assessment of bacterial susceptibility to antibiotics.

Antimicrobial Susceptibility Testing (AST) is a laboratory method used to identify effective antimicrobial agents for the treatment of individual patients and to predict treatment outcomes. The routine methods used for AST, which usually take 24 h for colony growth and a further 24 h for characterizing an isolate, are known as biochemical and phenotypic AST. The incubation period is typically overnight but can be longer for strains such as anaerobes, which need extra time for growth in a defined condition.

References

Suleiman M, Abu-Aqil G, Sharaha U, Riesenberg K, Lapidot I, Salman A, Huleihel M. Infra-red spectroscopy combined with machine learning algorithms enable early determination of Pseudomonas aeruginosa's susceptibility to antibiotics. Spectrochim Acta A Mol Biomol Spectrosc. 2022 Jun 5; 274:121080. doi: 10.1016/j.saa.2022.121080. Epub 2022 Feb 26. PMID: 35248858.

  1. Abu-Aqil, G., Sharaha, U., Suleiman, M., Riesenberg, K., Lapidot, I., Salman, A., Huleihel, M. (2022) Culture-independent susceptibility determination of E. coli isolated directly from patients’ urine using FTIR and machine-learning. Analyst 147, 4815, https://doi.org/10.1039/D2AN01253G

Jafari, M.J., Golabi, M., Ederth, T. (2024) Antimicrobial susceptibility testing using infrared attenuated total reflection (IR-ATR) spectroscopy to monitor metabolic activity, Spectrochimica Acta Part A: Molecular and Biomolecular Spectroscopy, Volume 304, 2024, 123384.

Comment 3: I think FT-MIR is a good technique that confirms the antibacterial of the extract but does not provide us with information related to the specific compound in the mixture that causes the inhibition.

Answer: The authors would like to express their gratitude for the suggestions and statements provided in the conclusion section, which will be further clarified. The antimicrobial effect of a specific extract is registered using FT-MIR techniques, relying on the spectral fingerprint of the cells. Detection of changes in the specific spectral region enables the determination of the antimicrobial effect of the compounds. With FT-MIR spectroscopy, the specific identification of antimicrobial compounds is not directly achieved. The antimicrobial effect of the compounds is obtained based on the biomolecular changes registered in the specific spectral region as a consequence of the action of the bioactive compounds. However, it's important to note that FT-IR spectroscopy, particularly with ongoing advancements, has the potential to differentiate between resistant microbial strains and non-resistant ones. Rather, MIR technology, coupled with chemometrics techniques, allows for the prediction of drug concentration in the assay. This prediction is based on specific compounds that have been previously isolated and characterized, utilizing partial least squares regression models. (Sampaio et al. 2020, Antibiotics).

The section was clarified and improved: (Yellow mark):

According to the promising results, the study suggests that the integration of FT-MIR spectroscopy and machine learning methods represents a promising strategy to register the cell effect of the compounds obtained from plant extracts and present antimicrobial properties. This holistic approach holds the potential for efficiently classifying and discovering new bioactive molecules, thereby streamlining the drug exploratory process. It was possible to develop a model based on the back-propagation network for plant extracts classification, containing active compounds based on the biomolecular characteristics recorded in the FT-MIR spectra of the bacteria.

Reviewer 2 Report

Comments and Suggestions for Authors

Few suggestions in order to improve the article: 

1. A scheme of the procedure, will make easier to understand the procedure, how is done the analysis, what is analysis by FT-MiR: the extract os the plant or the E. coli extract after treatment, etc. A figure with the procedure wull help the reader to understand the procedure.

2. It is hard to understand the group of compound that have been affected by the extracts, relation with the membrane, etc. A figure or ilustration will be helpful. 

3. Figure 7, only show how complicated it is but not give information, not detail. 

4. The natural compounds from the extract how you correlated with the effect E. coli, the molecules that you detect the effect? 

Author Response

Manuscript ID: antibiotics-2965304

Title: Enhancing Bioactive Compound Classification through the Synergy of Fourier-Transform Infrared Spectroscopy and Advanced Machine Learning Methods

Dear Editor,

We would like to thank you for the opportunity to improve and clarify the information present in the manuscript.  Then we resent, for your consideration, the revised version of our manuscript entitled “Enhancing Bioactive Compound Classification through the Synergy of Fourier-Transform Infrared Spectroscopy and Advanced Machine Learning Methods” that we would like to see published in the Special Issue Microbial Natural Products as a Source of Novel Antimicrobials (Antibiotics), that was significantly improved according to the referee’ suggestions. The manuscript was completely reviewed, and the main changes performed in the manuscript are highlighted in yellow marks.

I would like to inform you that the affiliation of the author Cecília Calado must be updated. Please, consider the following information:

4iBB – Institute for Bioengineering and Biosciences, i4HB – The Associate Laboratory Institute for Health and Bioeconomy, IST – Instituto Superior Técnico, Universidade de Lisboa, Av. Rovisco Pais, 1049-001 Lisbon, Portugal.

            I would like to inform you that all authors agree to publish this work in your journal.

Yours sincerely,

Pedro Sousa Sampaio

Reviewer 2 - Comments and Suggestions for Authors

We would like to thank you for the opportunity to improve and clarify the information present in the manuscript.  

Comment 1: A scheme of the procedure, will make it easier to understand the procedure, how is done the analysis, what is analysis by FT-MIR: the extract of the plant or the E. coli extract after treatment, etc. A figure with the procedure will help the reader to understand the procedure.

Answer: The authors would like to express their gratitude for your suggestion. A graphical representation of the experimental procedure was added to the manuscript (a new figure was added). The figure represents the different experimental procedures performed along the experimental work, from the extraction process to the classification of extracts according to the antimicrobial effects on E. coli cells based on the spectra registered using the spectrometer, and, after that, the spectral analysis.

Comment 2: It is hard to understand the group of compounds that have been affected by the extracts, their relationship with the membrane, etc. A figure or illustration will be helpful. 

Answer 2: Thank you for the suggestion. The action of the compounds in cellular biomolecules (membranes and organelles, among others) is recorded by MIR spectra as a consequence of variations in energetic variations. For a better framing of the main functional groups specific to each wavelength, a table was added that summarizes the main functional groups and the most important spectral regions for the interpretation of the work. The information was adapted from Sales et al. (2017). This detailed information represents the specific spectral region and the biochemical component that was changed/affected as a consequence of the compounds in the extracts.

Wavenumber (cm-1)

Vibrational mode in the functional group

Commonly assigned Biochemical Component

3300

ν(N-H)

Amide A: peptide, proteins

3100

ν(N-H)

Amide B: peptide, proteins

2960

νas(CH3)

Lipids

2929

2870

2850

νas(CH2)

νs(CH3)

νs(CH2)

Lipids

Lipids

Lipids

1740

ν(C=O)

Phospholipid esters

1655

[80% ν(C=O) and 20% ν(CN)] in O=C-N

Amide I in peptides and proteins

1550

ν (C-C) in O=C-N

Amide II in peptides and proteins

1450

νas(CH3)

Lipids, proteins

1395

νs(CH3)

Lipids, proteins

1380

νs(CH3)

Phospholipid, fatty acid, triglyceride

1240

νs(PO2-)

Phosphodiesters in nucleic acids, phospholipids and phosphorylates proteins

1160 and 1111

νs(C-O)

Ribose in RNA

1078

νs(PO2-)

Phosphodiesters in phosphate energetic level

1055

νs(C-O-P)

Phosphate esters

1032

def(C-OH)

Glycogen

965

νs(PO32-)

DNA and RNA

950

νs(PO32-)

Phosphorylated proteins

920

ν(COP)

Phosphorylated proteins

                          Vibrations type: νs , symmetric stretching; νas – asymmetric stretching ; def, deformations.

Comment 3: Figure 7 (8), only shows how complicated it is but does not give information, not detail. 

Answer 3: The authors would like to express their gratitude for your suggestion. The figure 8 represents the graphical schematization of the neuronal network developed for the identification of the antimicrobial and no-antimicrobial based on the spectral data obtained previously. Detailed information was added to the manuscript for a suitable explanation of figure 8:                      

The input layer represents the total 1154 absorbance spectra for different samples characterized by wavenumber registered by the spectrophotometer. The hidden layers are constituted each one by 20 nodes in each layer. The output layer represents the results in terms of the classification of the extracts (no-antimicrobial activity and antimicrobial activity).

Comment 4: The natural compounds from the extract how you correlate with the effect E. coli, the molecules that you detect the effect? 

Answer 4: Thank you for your question. In this work, a classic microbiological assay was previously carried out in which extracts obtained from different tissues were tested in a solid medium using E. coli cells. According to the results obtained (Fig. 1), some extracts exerted antimicrobial activity on E. coli cells according to the lower number of colonies obtained in some assay, comparatively with the control. Based on this information, the extracts were tested in the assay in which their antimicrobial effect was recorded in the MIR spectra, using E. coli cells for this purpose. Following the analysis of the experimental data using the PCA analysis, it was found that the samples tested with the extracts that showed more evident antimicrobial activity were resolved into different clusters, which could be characterized by having reduced metabolic activity as a result of the antimicrobial effect that may be present in the extracts. Based on these results, and in accordance with previous studies (Sampaio et al. 2020, Antibiotics), it can be assumed that the MIR spectroscopy technique does not specifically identify molecules with antimicrobial activity but highlights structural differences in terms of biomolecular composition recorded in near/medium infrared spectra. In the MIR spectra are registered the variations in the vibrational modes of the functional groups present in the main biomolecules as a consequence of the effect that the compounds exert on the biomolecules of the cells.
